# Diagnostics and Management of Pediatric Headache: An Exploratory Study among Dutch Physical Therapists

**DOI:** 10.3390/children10071135

**Published:** 2023-06-30

**Authors:** Maria N. Bot, Hedwig A. van der Meer, Marloes Meurs de Vries, Ewald M. Bronkhorst, Stanimira I. Kalaykova, Nico H. J. Creugers

**Affiliations:** 1Department of Dentistry, Radboud Institute for Health Sciences, Radboud University Medical Center, 6500 HB Nijmegen, The Netherlandsstanimira.kalaykova@radboudumc.nl (S.I.K.);; 2Academic Center for Dentistry Amsterdam, University of Amsterdam and Vrije Universiteit, 1081 LA Amsterdam, The Netherlands; h.a.vander.meer@acta.nl; 3The Royal Dutch Society for Physiotherapy, 3817 BA Amersfoort, The Netherlands; marloesmeurs@bcjunior.nl

**Keywords:** headache, pain, child, diagnostics, therapeutics, orofacial, temporomandibular disorders

## Abstract

Physiotherapists are often part of a multidisciplinary treatment plan for children with headaches. The literature on physical therapeutic diagnostics and management of headaches is often focused on adults. To gain insight, identify knowledge gaps, and increase the evidence needed for clinical physical therapeutic practice with children with headaches, an exploratory method is warranted. The purpose of this study was to describe the views, beliefs, and experiences of physical therapists regarding diagnostics and treatment options for children with headaches. The method consisted of a survey and two peer consultation group meetings. A total of 195 individual surveys were returned and 31 out of 47 peer consultation groups participated. Most participants were specialized in pediatric physical therapy (93.3%). They use the 4P-factor model (predisposing, precipitating, perpetuating, and protective factors) as a guiding principle in the diagnostic and therapeutic process in children with headaches. This model helps to organize and to understand how a variety of factors interact in a biopsychosocial relationship. Pediatric physical therapists focus their treatments on factors interfering with movement and functional abilities of the child with headaches. Knowledge of how temporomandibular disfunction can relate to headaches is currently insufficient for pediatric physical therapists.

## 1. Introduction

A headache is the second most common health complaint among children and adults and the second leading cause of disability worldwide [1,2]. Tension-type headaches (TTHs) and migraines are the most prevalent types of headaches among children and adults, with a global age standardized prevalence of 26.1% for TTHs and 14.4% for migraines. Headache prevalence rates increase with age [1,2]. Headache attacks can seriously affect quality of life, especially when they are frequent, and children with headaches often miss hours of school and avoid sports [3,4]. The percentage of years lived with disabilities (YLD) among children with headaches aged 5–14 years is 5%, and for youth aged 15 years or older it is 11.2% [1,2].

The different headache types and their diagnostic criteria are described in the International Classification of Headache Disorders (ICHD-3) [5]. Headache can be classified according to its clinical presentation and symptoms. In primary headache conditions, the specific etiology or cause cannot be determined. However, in the last few years, substantial knowledge has become available regarding pathophysiological mechanisms of primary headaches [5,6]. Peripheral and central sensitization of pain pathways has gained increasing attention for its role in chronic headache conditions. Headaches may become chronic as a consequence of central sensitization caused by prolonged nociceptive stimulation in myofascial tissues, and peripheral sensitization may induce a headache attack [5,7,8,9,10].

In secondary headaches, the headache is secondary to, or caused by, another medical condition, such as a disorder of the cervical spine or temporomandibular system [5,6]; for example, headaches attributed to temporomandibular disorder or a cervicogenic headache. Patients with these specific headache disorders are frequently seen by physical therapists.

A headache is a multifactorial condition, and a range of treatment options are available to manage pain and to decrease disabilities [11]. The physician, child, and parents decide together on the most appropriate treatment plan, including one or more treatment options for the individual situation. Pharmacological treatments, psychological treatments, or physical therapy treatments may all be part of a personalized multidisciplinary treatment plan [11]. For children with headaches, a variety of physical treatments are available, but most have not been studied. A recent systematic review and meta-analysis [12] reported an effectiveness of physical therapy treatments in a reduction on a pain index of 50% or more: Risk Ratio (RR) = 2.37 (95% Confidence Interval (CI) = 1.69 to 3.3312). The best evidence has been found for relaxation training, and when a headache is combined with temporomandibular disorders, occlusal appliances showed significant effects among children with headaches [12].

More high-quality evidence and future replication studies are needed in the field of physical therapy [12]. For this reason, not all guidelines specifically recommend physical therapy treatments for children with headaches [13,14]. This is a limiting factor in the treatment of children suffering from headaches when myofascial complaints and/or movement dysfunction are precipitating or perpetuating factors in the headache condition. This suggests that all kinds of treatment options in the field of physical therapy need to be studied.

The available literature on physical therapeutic diagnostic processes and physical examination for headache disorders is often focused on adults [15,16,17]. Although core outcome domains and measures regarding pediatric acute and chronic or recurrent pain for clinical trials are described [18], preferred patient-reported outcome measures (PROMs) for children with headaches in clinical physical therapy practice have not yet been established [18]. This makes it challenging for physical therapists who see children with headaches to follow evidence-based practice principles. Where there is a lack of evidence, the clinical experience of the physical therapist becomes more important in the diagnostic and management approach. To gain insight into the physical therapeutic clinical reasoning process for children with headaches, an exploratory method is warranted. By describing the views, beliefs, and experiences of physical therapists who treat children with headaches, knowledge and knowledge gaps can be identified for future research to focus on and increase the evidence needed for clinical practice. With this in mind, the purposes of this exploratory study were to describe the views, beliefs, and experiences of physical therapists about diagnostic and treatment options for children with headaches in the Netherlands.

## 2. Materials and Methods

### 2.1. Study Design

A two-part exploratory study consisting of a survey sent to individual participants (referred to as the ‘survey’) and two peer consultation group meetings (referred to as ‘focus groups’) was undertaken.

### 2.2. Study Participants

Of the thirteen physical therapy specializations in the Netherlands, the following treat children with headache: pediatric, manual, and orofacial physical therapists. All members of the associations of pediatric, manual, and orofacial physical therapists in the Netherlands (n = 2703) were therefore invited to participate in a survey about diagnostics and management of children with headaches. Invitations took the form of advertisements in the newsletters of each of the three associations. The survey was sent by mail to those physical therapists that agreed to participate in the study by responding to the advertisement both in January 2017 and March 2017.

Physical therapists in the Netherlands are often part of a peer consultation group. These groups consist of eight to twelve physical therapists who meet to discuss clinical cases and the latest scientific research three or four times a year. In 2017, invitations to participate in this study were sent to 47 peer consultation groups with a focus on pediatrics and/or headache in the Netherlands. Table 1 depicts the participant characteristics.

The Medical-Ethical Review Committee (METC) East Netherlands declared that the research does not fall under the Medical Research Involving Human Subjects Act (WMO), because the research participants are not subjected to WMO-compliant acts and no WMO-compliant behaviors are imposed on them (file number 2023-16543). The research was also submitted to the local review committee for non-WMO research (CMO) of Radboudumc. As no patients were involved, this study does not fall within the scope of the CMO Radboudumc. Therefore, the study implementation does not require a positive judgment from the CMO Radboudumc, the Eastern Netherlands METC, or another recognized medical ethics review committee.

### 2.3. Procedure

This study followed a five-step process, which is described below and depicted in Figure 1. The data collection was conducted between January 2017 and June 2019. The study used a ‘survey’ and ‘focus groups’ to collect data. Quantitative data were extracted from the survey, qualitative data were extracted from the focus groups, and consensus quotes from the focus groups were used to support the qualitative findings.

#### 2.3.1. Development of a Survey

First, a survey for this study was developed by the primary researcher, who is a pediatric, manual, and orofacial physical therapist, with the help of a pediatric neurologist. The survey consisted of questions about diagnostic screening (oral history items and anamnestic questionnaires), physical therapeutic clinical examination, and physical therapeutic management. A draft of the survey was compiled and sent to an expert group that consisted of clinicians practicing in the field of children with headaches (pediatricians; neurologists; pediatric, manual, and orofacial physical therapists; dentists specializing in orofacial pain; general practitioners; and the headache patient federation). This group was requested to provide opinions and help to further develop the survey. The survey was intended to gather various options, opinions, advice, and thoughts regarding the above-mentioned three clinical domains: diagnostic screening, physical therapeutic clinical examination, and physical therapeutic management. Modifications were made following the feedback received, and a final survey was established (Appendix A). After this, physical therapists were invited to participate in the survey part of this study.

#### 2.3.2. Survey

An invitation to participate in the survey part of this study was sent out as advertisements in the newsletters of pediatric, manual, and orofacial physical therapy associations. All participants individually returned their completed survey by email and ranked their agreement with each statement in the questionnaire. Participants scored their agreement or disagreement with statements on a 5-point Likert scale from 1 (“very useful”) to 5 (“not useful”). Each participant wrote down their opinions by answering open questions. Demographic information and information regarding the work environment were also collected.

After they had returned the completed survey, participants were asked to join the focus group part of the study with their existing peer consultation group, or to join one of the participating groups.

#### 2.3.3. Focus Groups

Forty-seven existing peer consultation groups (comprising 8–12 physical therapists) with a focus on pediatrics and/or headaches were asked to participate in the study. Members of the participating groups who had not completed the survey before attending the focus group were invited to complete the survey at this stage. Participation in the survey part of the study was not a requirement for participating in the focus group, which meant that new information might be raised in the focus group discussions by participants who had not completed the survey.

Subsequently, two focus group meetings were held to provide all individual participants with an opportunity to have their opinions considered by other group members and to generate group views, ideas, and recommendations for the authors.

All participating groups received a detailed review of the literature as background material and PowerPoint presentations to give guidance about what to focus on in the two focus group meetings. During the two meetings, individual and group ideas, opinions, and recommendations were gathered. The results were summarized and reported for each meeting by a member of the focus group acting as a contact person for the research group.

The group discussion focused on a few questions aimed at gaining knowledge, views, and beliefs of the participants on primary headaches, secondary headaches, headaches in relation to biopsychosocial factors, and headaches in relation to co-morbid musculoskeletal complaints in the temporomandibular and neck region. Participants were free to give broad answers related to these topics in order to avoid a narrow focus based on the questions asked.

The following questions were used for group discussion:‘How can biopsychosocial screening be performed, and which anamnestic questionnaires can be used?’‘What are your thoughts on a relationship between headache and musculoskeletal pain, and dysfunction in the temporomandibular and neck region? Should you examine musculoskeletal pain and dysfunction in the temporomandibular and neck region of a child with headache in the physical therapy practice? Are all physical therapists competent to perform this examination?’‘Do you think there is an association between headache and head/neck position? If so, how do you examine and treat head/neck position?’‘What are the treatment goals for physical therapists in general and what does your team of practitioners look like (mono- or multidisciplinary)?’

### 2.4. Data Analysis

Data from the completed questionnaires were entered in a spreadsheet. Frequency tables and histograms were created and analyzed. For this, we used Microsoft Excel version 2016 and the Statistical Package for the Social Sciences (IBM SPSS) version 25.0 (SPSS Corp., Chicago, IL, USA).

## 3. Results

### 3.1. Study Participants

A total of 195 individual surveys were returned and all participants of the individual round joined the focus group meetings. The majority of participants in the survey were females (97.4%), specialized in pediatric physical therapy (93.3%), and worked in a physical therapy practice (96.9%).

A total of 31 (65.9%) out of 47 physical therapy peer consultation groups with a focus on pediatrics and/or headaches participated in the study focus group meetings, of which almost two-thirds also participated in the survey.

The participants reported a total of 620 patients with headache attacks once or more per month (3 to 4 per physical therapist), amongst them 319 children (1 to 2 per physical therapist). Most participants of the study treated children with (multiple) chronic pain conditions, including headaches. A total of 37 (19%) physical therapists used ICHD-3 criteria for headaches [5].

### 3.2. Oral History

#### 3.2.1. Survey

Across all anamnestic topics, the percentage of participants that considered a specific topic (very) useful varied between 45% and 98%. Most participants found it useful to ask their patient if daily activities increased their headaches (98%). Other useful questions were aimed at gaining information about the severity (97%), location (94%), duration (93%), and frequency (93%) of headaches and about stress (96%). The questions that the least number of participants found useful to ask were about temporomandibular disorders (53%) or pain/sensitivity in the teeth (55%). Figure 2 depicts all incorporated anamnestic items of the survey and their percentages of usefulness.

#### 3.2.2. Focus Groups

All physical therapists considered it important to work with the International Classification of Functioning, Disability, and Health for Children and Youth (ICF-CY) [19] and found the 4P-factor model (predisposing, precipitating, perpetuating, and protective factors) to be very helpful [20,21,22] (see Appendix A). The majority of participants were unfamiliar with temporomandibular disorders, pain in the teeth, and bruxism. Owing to a lack of knowledge, they were not used to asking the child with headaches questions about these specific topics. Participants did not explicitly focus on musculoskeletal pain and dysfunction in the temporomandibular and neck region as manual physical therapists and orofacial physical therapists would in children with headaches.

The following consensus quote was obtained:

“Manual physical therapists might be better in classifying headache according to the ICHD-3 and pediatric physical therapists are better in classifying according to the ICF-CY”.

### 3.3. Anamnestic Questionnaires

#### 3.3.1. Survey

Headache diaries were used by 92.5% of the participants. A quarter of the participants (26%) described which anamnestic questionnaires they used in their practice for children with headaches. Most used were the Patient Specific Functional Scale [23] Dutch version (PSK) [24] at 52%; the Headache Impact Test 6-items (HIT 6) [25] at 28%; and the Pediatric Migraine Disability Assessment Questionnaire (Pedmidas) [26] at 24%. Other questionnaires mentioned (n = 27) were not headache-specific. All anamnestic items incorporated in the survey and their percentages of usefulness are depicted in Figure 2.

All anamnestic questionnaires mentioned and their frequency of use are set out in Table 2.

#### 3.3.2. Focus Groups

Anamnestic questionnaires were considered helpful as an addition to oral history. A headache diary was considered useful to classify headaches, or to obtain a better understanding of the headache complaints and their predisposing, precipitating, perpetuating, and protective factors. Keeping up a headache diary for one week might already be enough for this, depending on the frequency of the headaches. Physical therapists advised using a headache diary when the oral history was not sufficient to classify the headache condition.

The following consensus quote was obtained:

“We find it’s important not to focus on pain. By keeping up a headache diary, pain gets full attention on a daily basis”.

### 3.4. Physical Therapeutic Clinical Examination

#### 3.4.1. Survey

The items considered to be most useful to examine (>90%) by the participants were neck, muscle palpations, and head posture. The items considered less useful to examine (>70%) were intraoral inspection, body mass index, and measuring the size of the head. All clinical examination items incorporated in the survey and their percentages of usefulness are depicted in Figure 3.

#### 3.4.2. Focus Groups

It was considered important to distinguish between a headache and neck and orofacial pain and disorders. Moreover, most participants were uncertain whether they were able to examine neck and temporomandibular pain and disorders, especially the movement examination of the jaw (>50%). Important items that were missing from the examination list of the survey were cranial nerve examination, physical effort tolerance, breathing techniques, thoracic spine examination, and sleep posture. All therapists considered it necessary to observe task-related head posture in sitting and standing positions. The participants mentioned that pictures could be taken by therapists or parents in order to measure the posture and make it visible to the patient. The participants considered that a misaligned posture could lead to muscle overload and increased pressure on the cervical spine with a headache in response. In addition to observing posture, participants mentioned the Cranio-Cervical Flexion Test (CCFT) [27] and Deep Neck Flexor Endurance Test (DNFET) [28] as valuable instruments for children with headaches.

The following consensus quote was obtained:

“As pediatric physical therapists, we must identify whether the jaw region and/or neck region influence the child’s headache symptoms, but we do not know how to examine the jaw region and are not that familiar with the neck region just as manual therapists”.

### 3.5. Physical Therapeutic Management

#### 3.5.1. Survey

The treatment items considered most useful in a physical therapeutic setting by 90% or more of the participants were headache and pain education, self-monitoring advice, lifestyle coaching, sleep hygiene advice, exercise training, posture training, and relaxation training.

The treatment items considered least useful were manipulations of the neck (33%) and dry needling (20%), and items considered unknown in a physical therapeutic setting were provision of occlusal appliances (69%), hypnosis (67%), and meditations (58%). The usefulness of the physical therapy treatment items by percentage is presented in Figure 4.

#### 3.5.2. Focus Groups

The participants stated that treating children with headaches was complex owing to the variety of precipitating and perpetuating factors, and therefore there was no one ideal physical therapy treatment option that could be given to all children with headaches. Participants often used an active approach that consisted of graded exercise, return to activity, and graded exposure. They also explained the underlying biological mechanisms of pain to the child and parents.

Physical therapists expressed their willingness to be a part of a multidisciplinary team and to refer to other specialists when necessary. According to the physical therapists that participated in the focus groups, a school coordinator or teacher should be part of this team. Participants also mentioned that they did not always know what services other specialized physical therapists (e.g., manual, pediatric or orofacial physical therapists) could offer a child with headaches.

The following consensus quote was obtained:

“If we suspect a relation of headache with neck dysfunction, we will refer to a manual physical therapist. For dysfunction in the mouth area, we will refer to a speech therapist”.

## 4. Discussion

This article describes the first study to explore and summarize the opinions of physical therapy practitioners in the Netherlands regarding the diagnostics and management provided for children with headaches. Based on the results of the study, recommendations are made for innovative diagnostic and management approaches.

### 4.1. Diagnostics

Diagnostic screening was considered the most important in classifying headache conditions. A headache diary and anamnestic questionnaires (Pedmidas and HIT 6) are considered helpful additions to the oral history to classify the headache condition. This is in line with the described core outcome domains and measures for pediatric acute and chronic or recurrent pain for clinical trials in the literature [18]. The study of McGrath et al. recommends the use of a headache and sleep diary. However, participants in this study were of the opinion that it is important not to focus on the pain, which might be enhanced by keeping up a headache diary on a daily basis. Allowing pain to take center stage may cause and increase the activation of the pain neuromatrix [29]. Practitioners must therefore always carefully consider when, for how long, and with whom to use a headache diary.

The participants in our study considered it important to gather all available diagnostic information of a child with headaches and put this into a biopsychosocial model of the ICF-CY [19]. The ICF-CY provides both a detailed classification of aspects of children’s health and function and a pictorial framework that brings all heath issues within a broader biopsychosocial context together. It could be considered to expand the scope of the ICF-CY framework by adapting the F-words (functioning, family, fitness, fun, friends, and future) [30,31]. F-words can be used to operationalize the ICF-CY, support a holistic approach to childhood disability, and inform physical-activity- and rehabilitation-based interventions in a light that is focused on ‘can do’ rather than ‘cannot do’ [30,31]. Tools and resources are freely available on CanChild’s F-words Knowledge Hub (CanChild, 2019) www.canchild.ca/f-words (accessed on 22 June 2023)).

The participants in our study found it important to address predisposing, precipitating, perpetuating, and protective factors (4P) [20,21,22]. Using the 4P-factor model during the diagnostic process, a physical therapist can organize risk and protective factors with regard to the biopsychosocial model of the ICF-CY [19,20,21,22]. This model helps to understand how a variety of factors may interact with the headache condition without any hierarchy of implied importance. This suggests that changes in any area of the 4P-factor model may potentially have influences elsewhere in the system. Physical therapists and other healthcare providers might consider using both the F-words within the ICF-CY framework and the 4P-factor model for a more holistic approach on all aspects of children’s health and function interacting with the headache condition.

Physical therapists focus on health problems that affect movement and functional abilities [32]. In carrying out processes of clinical reasoning, physical therapists may need to collaborate with other professionals for additional information [32]. Examination items in our study consisted of generic and region-specific movement and functional abilities, which were considered important by all participants of this study. The participants also found it important to examine neck and temporomandibular pain and dysfunction because of their association with headache disorders [33,34,35,36]. However, most pediatric physical therapists in our study reported not being adequately trained regarding examining neck and temporomandibular pain and dysfunction and being less familiar with this examination than manual therapists and orofacial physical therapists would be.

While there is no expert consensus in the literature regarding physical examination of neck pain among children with headaches, consensus exists for adults [17]. The recommended musculoskeletal examination items for adults with headaches are as follows: manual joint palpation, the cervical flexion rotation test, active range of cervical movement, head forward position, trigger point palpation, muscle tests of the shoulder girdle, passive physiological intervertebral movements, reproduction and resolution of headache symptoms, screening of the thoracic spine, and combined movement tests [17]. Some of these tests (active and passive range of cervical movement, head forward position, flexion rotation test, and muscle palpation) have also been used in studies with children with headaches to explore the mechanical dysfunction of the cervical spine [37,38].

To examine temporomandibular pain and dysfunction among children and adolescents, an international consensus study was recently published by Rongo et al. [39]. This Delphi study developed new instruments and adapted the diagnostic criteria for temporomandibular disorders (DC/TMD) for the evaluation of temporomandibular disorders in children and adolescents [39]. To improve physical therapists’ awareness of temporomandibular symptoms, three screening questions (3Q/TMD) are recommended by Rongo et al. [39]:‘Do you have pain in your temple, face, jaw or jaw joint once a week or more?’‘Do you have pain once a week or more when you open your mouth or chew?’‘Does your jaw lock or become stuck once a week or more?’

As most physical therapists in our study stated that they did not have the knowledge to determine whether temporomandibular signs and symptoms are present, these screening questions could be useful for them. Physical therapists who suspect the presence of a painful temporomandibular disorder in children with headaches after the use of these screening questions could refer them to an orofacial pain specialist (dentist or physical therapist).

### 4.2. Management

Evidence is lacking as to whether a pharmacological or non-pharmacological intervention is the best approach to use with children with headaches [38,39]. The latest literature on the management of children with headaches encourages a multidisciplinary approach involving physical therapy [40,41]. Physical therapy can be a helpful discipline for headache management among children, with the intervention focusing on a more active approach using graded exposure, graded exercise, return to activity, and daily aerobic exercises [11,12]. The primary aim of physical therapy in treating headache conditions is to reduce the impact of pain on daily activities and quality of life and to enable the patient to cope better with headache complaints in school, sports, and other settings [11,12]. A physical therapist offers tailor-made support in stimulating, rediscovering, retaining, and/or optimizing movement and functional abilities of the child with headaches [32]. By using both the F-words within the ICF-CY framework and the 4P-factor model, healthcare professionals including physical therapists provide a basis for a patient-centered multidisciplinary treatment approach for children with headaches and their parents. The healthcare professional can explain the individual headache condition in terms of its predisposing, precipitating, perpetuating, and protective factors within the 4P-factor model [20,21,22]. A variety of factors may interact with the headache condition and level of functioning, of which some factors can be addressed and others cannot. Based on the child and parents’ personal values and preferences, treatment goals can easily be set and implemented in a multidisciplinary manner by using the 4P-factor model [20,21,22]. Use of the 4P-factor model will enhance communication and coordination among the child, parents, and healthcare professionals, such as pediatricians, neurologists, psychologists, physical therapists, optometrists, and dentists. Within a multidisciplinary treatment, physical therapy treatment goals focus on 4P-factors that interfere with the movement and functional abilities of the child with a headache and incorporate the F-words (function, family, fitness, fun, friends, and future) in the ICF-CY to support a holistic approach to childhood disability [20,21,22,30,31].

Graded exercise, return to activity, and graded exposure were commonly used by the physical therapists in our study. These active treatments incorporate behavioral and cognitive approaches to improve activity tolerance in sports and functional activities despite headache complaints [11,12]. Physical therapists in our study combined this active approach with explaining the underlying biological mechanisms of pain to the child and parents. They also applied treatment strategies to certain biological precipitating and perpetuating factors in lifestyle and physical abilities in children with headaches. Depending on the individual headache sufferer, participants used lifestyle coaching, sleep hygiene advice, self-monitoring advice, exercise treatment, relaxation training, habit-reversal techniques, mobilization, massage and stretching techniques, and posture training. In the literature, lifestyle changes such as sleep deprivation, missing meals (particularly breakfast), and busy schedules are often cited as precipitating factors for headache complaints [40]. Another effective treatment option for children with headaches is relaxation training [12], and exercises can be prescribed to reduce pain and improve quality of life for both short- and long-term follow-up compared to the usual care for children with chronic pain [11,42]. Remotely delivered physical therapy treatments might be potentially useful and cost effective for children with headache [43].

In general, participants of our study, most of whom were pediatric physical therapists, were of the opinion that neck pain and dysfunction can be better treated by manual physical therapists and temporomandibular pain and dysfunction can be better treated by orofacial pain specialists (physical therapists or dentists). In children with headaches, the treatment of co-morbid temporomandibular disorders is important, as orofacial physical therapy combined with occlusal appliances is effective for children with headaches and co-morbid temporomandibular disorders [12,44]. It is important for physical therapists to know common co-morbid disorders, how to screen for them, and who to collaborate with to optimize the child’s treatment plan. This may require interprofessional courses or discussions within a clinical setting.

### 4.3. Limitations

Our results suggest that physical therapists who see children with headaches in the Netherlands are mainly pediatric physical therapists (93.3%). It is possible that this study missed manual physical therapists and/or orofacial physical therapists with a special interest in children with headaches, owing to a lack of involvement in their peer consultation groups. Representativeness of the participants could not be ensured because inclusion depended on the willingness of the participants. It is likely that only highly engaged physical therapists responded.

Another limitation is that most pediatric physical therapists treat children with (multiple) chronic pain conditions but are not specifically experts on temporomandibular disorders and headaches.

## 5. Conclusions

In the physical therapeutic care of children with headaches, participants in the study used the ICF-CY and 4P-factor model during the diagnostic process. A personalized multidisciplinary treatment plan can be created based on the findings and these models. F-words can be used to operationalize the ICF-CY and to support a holistic approach to childhood disability. Graded exercise, return to activity and graded exposure, headache and pain education, self-monitoring advice, lifestyle coaching, sleep hygiene advice, exercise training, posture training, and relaxation training were all considered useful treatment options for children with headaches.

Predisposing, precipitating, or perpetuating factors in a child with headaches that need to be considered are neck pain and dysfunction as well as temporomandibular pain and dysfunction. However, participants stated that their knowledge of these subjects was insufficient to apply them in their routine with these patients. This suggests that more education is needed among pediatric physical therapists to enable them to apply these factors in examining and treating their patients, and to refer these patients earlier to a specialized colleague when necessary.

## Figures and Tables

**Figure 1 children-10-01135-f001:**
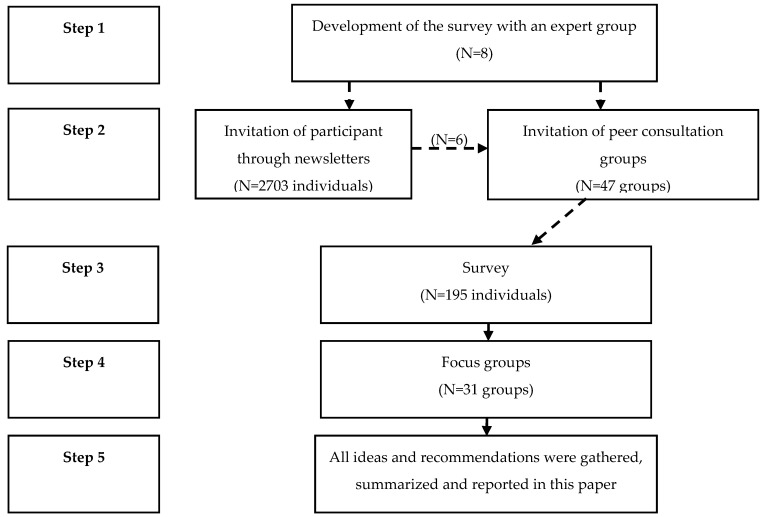
Steps of the study.

**Figure 2 children-10-01135-f002:**
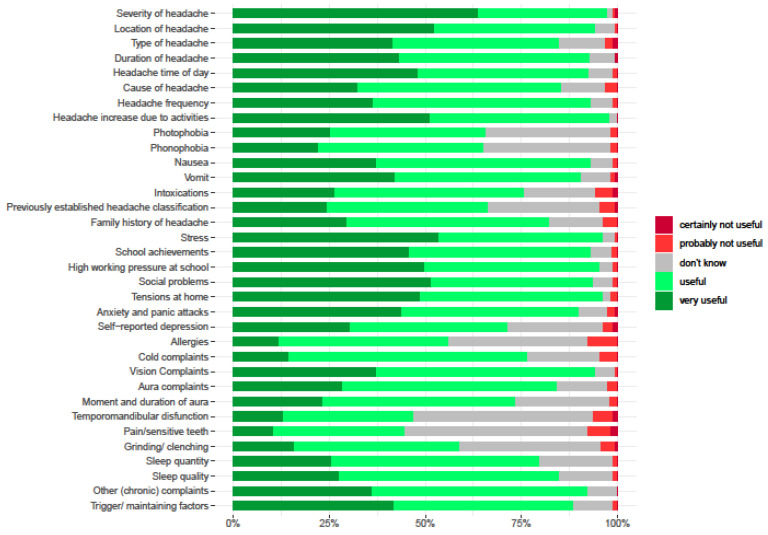
Anamnestic items and percentages useful or not useful of participants.

**Figure 3 children-10-01135-f003:**
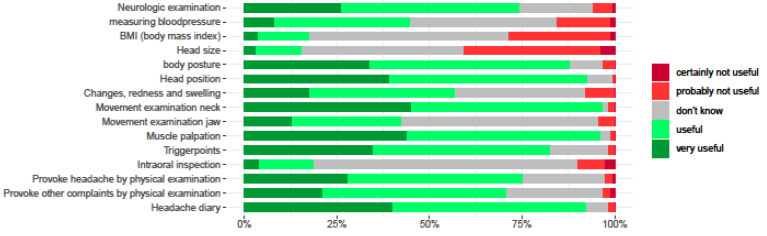
Clinical examination items and percentages useful or not useful of participants.

**Figure 4 children-10-01135-f004:**
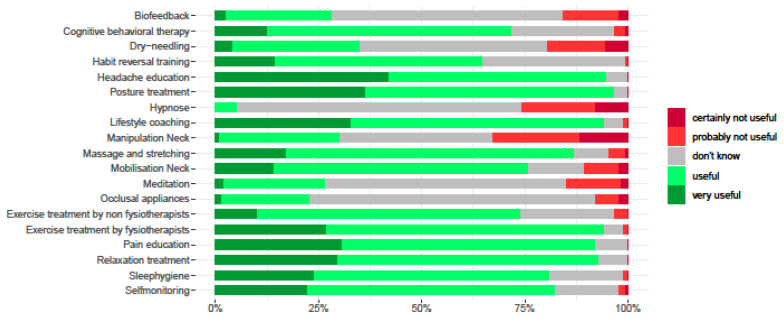
Management items and percentages useful or not useful of participants.

**Table 1 children-10-01135-t001:** Participant characteristics.

Characteristic	Percentage	Number ^1^
**Sex**		
Male	2.6%	5
Female	97.4%	190
Total	100.0%	195
**Age**		
25–29 Age (y)	6.2%	12
30–39 Age (y)	37.4%	73
40–49 Age (y)	23.6%	46
50–59 Age (y)	21.0%	41
60+ Age (y)	11.8%	23
**Academic degree**		
Bachelor	85.6%	167
Master	14.4%	28
**Specialization in physical therapy**		
Physical therapist	100.0%	195
Pediatric specialist	93.3%	182
Manual specialist	2.6%	5
Orofacial specialist	2.1%	4
**Workplace**		
University hospital	1.0%	2
Regional hospital	4.6%	9
Rehabilitation	5.6%	11
First-line health centre	12.3%	24
Physical therapy practice	84.6%	165
Other	7.2%	14
**Number of patients**		
Total number of patients with headache attacks at least once per month seen by focus group participants		620
Total number of children with headache attacks at least once per month seen by focus group participants		319
**Classification**		
Use of ICHD-3 criteria	19.0%	37

^1^ Number varies for each variable due to missing data.

**Table 2 children-10-01135-t002:** Anamnestic questionnaires and frequency of use.

Headache Specific Questionnaires	Percentage of 50 Participants
Patient Specific Complaints with a Numeric (Pain) Rating Scale 0-11 (NRS/NPRS)	52%
Headache Impact Test 6-items (HIT 6)	28%
Pediatric Migraine Disability Assessment Questionnaire (Pedmidas)	24%
Headache Disability Index (HDI)	2%
**Quality of Life (QoL) questionnaires**	
Pediatric Quality of Life Inventory (PedsQL)	10%
Short version; Dutch Children TNO-AZL Quality of Life Questionnaire (DUX-25)	6%
Kidscreen	2%
**Other questionnaires**	
Four-Dimensional Symptom Questionnaire (4DSQ) (distress, depression, anxiety and somatization)	12%
Pain Coping Inventory (PCI)	12%
Neck Disability Index (NDI)	6%
Multidimensional Fatigue Inventory (MFI)	6%
Brief Illness Perception Questionnaire (BIPQ)	4%
Central Sensitization Inventory (CSI)	2%
Pain Coping and Cognition List (PCCL)	2%
Tampa scale of kinesiophobia (TSK)	2%
Fatigue Severity Scale (FSS)	2%
Children’s Nonverbal Learning Disabilities Scale (C-NLD)	2%

## Data Availability

The data from the survey are available as Appendix A. Data from the focus groups are available on request from the corresponding author. These data are not publicly available due to privacy of participants.

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
