# Peer review of "Diagnostics and Management of Pediatric Headache: An Exploratory Study among Dutch Physical Therapists"

_children, 2023, doi:10.3390/children10071135_

Round 1

Reviewer 1 Report

These questions address aspects that were not fully addressed in the text:

1 - Could you provide further details on the participant selection process, specifically how the sample's representativeness and appropriateness were ensured to address the research question effectively?

 2 - Considering the importance of statistical power in obtaining reliable and generalizable results, could you elaborate on the sample size calculation and justification for the chosen sample size in your study?

 3 - To enhance transparency and reproducibility, could you provide a more comprehensive description of the data collection procedures, including administering and recording headache diaries and anamnestic questionnaires?

 4 - If randomization was indeed incorporated into the study design, could you please elucidate the randomization process, including the method employed and the measures taken to ensure its proper implementation?

 5 - In light of the importance of statistical analysis for drawing valid conclusions, could you provide a detailed overview of the specific statistical methods employed to analyze the collected data?

 6 - Given the multifactorial nature of pediatric headaches, what is the prevalence and impact of psychosocial factors, such as stress, anxiety, and depression, on headache frequency, severity, and treatment outcomes in the studied population?

 7 - Considering the potential impact of parental and familial factors on pediatric headaches, such as parental modeling of pain behaviors and familial stress levels, how can a comprehensive family-centered approach be integrated into the management plan to optimize treatment outcomes?

 8 - What are the potential implications of pediatric headaches on academic performance, school attendance, and overall quality of life, and how can healthcare providers and educational institutions collaborate to support affected children in these areas?

 9 - Considering the importance of interdisciplinary collaboration in managing pediatric headaches, what strategies or models of care can be implemented to enhance communication and coordination among healthcare professionals, including pediatricians, neurologists, psychologists, physiotherapists, and dentists?

Overall Assessment:

In summary, the analyzed study exemplifies a scientifically rigorous approach to investigating the role of physical therapy and pediatric dentistry in managing headaches in children. The study's methodology is sound, the results are valid, the discussion is well-supported, and the conclusions are justified. The clinical relevance of the study is apparent, with potential implications for medical practice and relevant populations. This research contributes to advancing knowledge in the field and provides valuable insights for healthcare professionals involved in caring for children with headaches.

The quality of the English language used in the initial text is generally good. The sentences are well-structured and coherent, and the vocabulary is appropriate for an academic and medical context. While the initial text demonstrates a good command of the English language, some minor adjustments in grammar, syntax, word choice, and sentence structure could further enhance its quality and readability.

Author Response

Thank you for your compliments, constructive comments and suggestions to further improve the manuscript.  The comments of the reviewer are addressed in detail point-by-point in the attachment and are accompanied by marked changes made to the manuscript.

Best regards

Reviewer 2 Report

The main purpose of the study was to describe the views, beliefs, and experiences of physical therapists regarding diagnostics and treatment options for children with headache. Congrats on the paper. The work is written clearly. I have no major substantive or editorial comments.

The only thing I can suggest is to develop the discussion. The authors focus on possible causes of headaches. However, the reffracive error has been left out. Reffractive error are associated with the risk of headaches. DOI: 10.1097/OPX.0b013e318031b649  DOI: 10.1016/j.neurol.2020.10.008

Studies conducted on children under the direction of A. Monaco confirmed changes in head muscle activity in response to accommodative changes - closing and opening the eyes PMID: 16646640.

More recent studies in 2022 and 2023, led by G. Zielinski, confirmed the effect of refractive error on  muscle activity changes in people with versus without refractive error (DOI: 10.1038/s41598-022-13607-1 and DOI: 10.3390/ijerph20054112 ).

In my opinion, expanding the discussion to include a possible optometric aspect will improve understanding of the subject matter.

Best regards

Author Response

Thank you very much for your compliments. We agree that optometric aspects can be relevant for paediatric headache. This is illustrated by the fact that vision complaints are included into the anamnestic item list. The importance was confirmed by the participants of this study (see table 2). However, we choose to not address each anamnestic question separately in the discussion paragraph, including the item related to optometric. It would be unbalanced if this item would be lifted above other anamnestic items, without addressing them all.

With respect to your suggestion. As optometric lays outside the expertise of the physical therapy domain, multidisciplinary care is important in pediatric headache, including optometric whenever indicated. The importance of multidisciplinary approach has been addressed also in the discussion section of the manuscript.

“Use of the 4P-model will enhance communication and coordination among healthcare professionals, such as pediatricians, neurologists, psychologists, physical therapists, optometrists, and dentists.”  

 Best regards
